# Walking the Tightrope: An Investigation of the Convolutional Autoencoder Bottleneck

## Abstract

In this paper, we present an in-depth investigation of the convolutional autoencoder (CAE) bottleneck. Autoencoders (AE), and especially their convolutional variants, play a vital role in the current deep learning toolbox. Researchers and practitioners employ CAEs for various tasks, ranging from outlier detection and compression to transfer and representation learning. Despite their widespread adoption, we have limited insight into how the bottleneck shape impacts the CAE's emergent properties. We demonstrate that increased bottleneck area (i.e., height × width) drastically improves generalization in terms of reconstruction error while also speeding up training. The number of channels in the bottleneck, on the other hand, is of secondary importance. Furthermore, we show empirically that CAEs do not learn an identity mapping, even when all layers have the same number of neurons as there are pixels in the input. Besides raising important questions for further research, our findings are directly applicable to two of the most common use-cases for CAEs: In image compression, it is advantageous to increase the feature map size in the bottleneck as this improves reconstruction quality greatly. For reconstruction-based outlier detection, we recommend decreasing the feature map size so that out-of-distribution samples will yield a higher reconstruction error.

## 1 Introduction

Autoencoders (AE) are an integral part of the neural network toolkit. They are a class of neural networks that consist of an encoder and decoder part and are trained by reconstructing datapoints after encoding them. Due to their conceptual simplicity, autoencoders often appear in teaching materials as introductory models to the field of unsupervised deep learning. Nevertheless, autoencoders have enabled major contributions in the application and research of the field. The main areas of application include outlier detection Xia et al. (2015); Chen et al. (2017); Zhou & Paffenroth (2017); Baur et al. (2019), data compression Yildirim et al. (2018); Cheng et al. (2018); Dumas et al. (2018), and image enhancement Mao et al. (2016); Lore et al. (2017). Additionally, autoencoders can be used as catalysts in the training of deep neural networks. The layers of the target network can be greedily pre-trained by treating them as autoencoders with one hidden layer Bengio et al. (2007). Subsequently, Erhan et al. (2009) demonstrated that autoencoder pre-training also benefits generalization. Currently, researchers in the field of representation learning frequently rely on autoencoders for learning nuanced and high-level representations of data Kingma & Welling (2013); Tretschk et al. (2019); Shu et al. (2018); Makhzani et al. (2015); Berthelot et al. (2018).

However, despite its widespread use, we propose that the (deep) autoencoder model is not well understood. Many papers have aimed to deepen our understanding of the autoencoder through theoretical analysis Nguyen et al. (2018); Arora et al. (2013); Baldi (2012); Alain & Bengio (2012). While such analyses provide valuable theoretical insight, there is a significant discrepancy between the theoretical frameworks and actual behavior of autoencoders in practice, mainly due to the assumptions made (e.g., weight tying, infinite depth) or the simplicity of the models under study. Others have approached this issue from a more experimental angle Arpit et al. (2015); Bengio et al. (2013a); Le (2013); Vincent et al. (2008); Berthelot et al. (2019); Radhakrishnan et al. (2018). Such investigations are part of an ongoing effort to understand the behavior of autoencoders in a variety of settings.

The focus of most such investigations so far has been the traditional autoencoder setting with fully connected layers. When working with image data, however, the default choice is to use convolutions,

as they provide a prior that is well suited to this type of data Ulyanov et al. (2018). For this reason, Masci et al. (2011) introduced the convolutional autoencoder (CAE) by replacing the fully connected layers in the classical AE with convolutions. In an autoencoder, the layer with the least amount of neurons is referred to as a bottleneck. In the regular AE, this bottleneck is simply a vector (rank-1 tensor). In CAEs, however, the bottleneck assumes the shape of a multichannel image (rank-3 tensor, height $\times$ width $\times$ channels) instead. This bottleneck shape prompts the question: What is the relative importance of bottleneck depth (i.e., the number of channels) versus the bottleneck area (i.e., feature map size) in determining the tightness of the CAE bottleneck? Intuitively, we might expect that only the total number of neurons should matter since convolutions with one-hot filters can distribute values across channels.

In this paper, we share new insights into the properties of convolutional autoencoders, which we gained through extensive experimentation. We address the following questions:

- How do bottleneck area and depth impact
  - reconstruction quality?
  - generalization ability?
  - knowledge transfer to downstream tasks?
- How and when do CAEs overfit?
- Are CAEs capable of learning a "copy function" if the CAE is complete (i. e., when the number of pixels in input equals the number of neurons in bottleneck)? By copy function we are referring to a type of identity function, in which the input pixel values are transported through the bottleneck and copied to the output. The hypothesis that AEs learn an identity mapping is common for fully connected AEs and can sometimes be encountered for CAEs (see Sections 4 and 5 in Masci et al. (2011).

We begin the following section by formally introducing convolutional autoencoders and explaining the convolutional autoencoder model we used in our experiments. Additionally, we introduce our three datasets and the motivation for choosing them. In Section 3, we outline the experiments and their respective aims. Afterward, we present and discuss our findings in Section 4. All of our code, results, and trained models and datasets, is published on github. We invite interested readers to take a look and experiment with our models.

## 2 MATERIALS AND METHODS

### 2.1 AUTOENCODERS AND CONVOLUTIONAL AUTOENCODERS

The regular autoencoder, as introduced by Rumelhart et al. (1985), is a neural network that learns a mapping from data points in the input space $\boldsymbol{x} \in \mathbb{R}^d$ to a code vector in latent space $\boldsymbol{h} \in \mathbb{R}^m$ and back. Typically, unless we introduce some other constraint, $m$ is set to be smaller than $d$ to force the autoencoder to learn higher-level abstractions by having to compress the data. In this context, the encoder is the mapping $f(\boldsymbol{x}) : \mathbb{R}^d \to \mathbb{R}^m$ and the decoder is the mapping $g(\boldsymbol{h}) : \mathbb{R}^m \to \mathbb{R}^d$. The layers in both the encoder and decoder are fully connected:

$$\boldsymbol{l}^{i+1} = \sigma(\boldsymbol{W}^i \boldsymbol{l}^i + \boldsymbol{b}^i). \tag{1}$$

Here, $\boldsymbol{l}^i$ is the activation vector in the i-th layer, $\boldsymbol{W}^i$ and $\boldsymbol{b}^i$ are the trainable weights and $\sigma$ is an element-wise non-linear activation function. If necessary, we can tie weights in the encoder to the ones in the decoder such that $\boldsymbol{W}^i = (\boldsymbol{W}^{n-i})^T$, where $n$ is the total number of layers. Literature refers to autoencoders with this type of encoder-decoder relation as weight-tied.

The convolutional autoencoder keeps the overall structure of the traditional autoencoder but replaces the fully connected layers with convolutions:

$$\mathsf{L}^{i+1} = \sigma(\mathsf{W}^i * \mathsf{L}^i + \boldsymbol{b}^i), \tag{2}$$

where $*$ denotes the convolution operation and the bias $\boldsymbol{b}^i$ is broadcast to match the shape of $\mathsf{L}^i$ such that the j-th entry in $\boldsymbol{b}^i$ is added to the j-th channel in $\mathsf{L}^i$. Whereas before the hidden code was an m-dimensional vector, it is now a tensor with a rank equal to the input tensor's rank. In the case

of images, that rank is three (height, width, and the number of channels). CAEs generally include pooling layers or convolutions with strides $> 1$ or dilation $> 1$ in the encoder to reduce the size of the input. In the decoder, unpooling or transposed convolution layers Dumoulin & Visin (2016) inflate the latent code to the size of the input.

## 2.2 OUR MODEL

Our model consists of five strided convolution layers in the encoder and five up-sampling convolution layers (bilinear up-sampling followed by padded convolution) Odena et al. (2016) in the decoder. We chose to use five such layers so that the bottleneck area would be 3x3 (input dimensions are $96 \times 96$). To have a deeper model, we added two residual blocks He et al. (2016) with two convolutions each after each strided / up-sampling convolution layer. We applied instance normalization Ulyanov et al. (2016) and ReLU activation Nair & Hinton (2010) following every convolution in the architecture.

Our goal was to understand the effect latent code shape has on different aspects of the network. Therefore, we wanted to change the shape of the bottleneck between experiments while keeping the rest of the network constant. To this end, we quadrupled the number of channels with every strided convolution $s^i$ and reduced it by a factor of four with every up-sampling convolution $u^i$. In effect, this means that the volume (i. e., height $\times$ width $\times$ channels) of the activations is identical to that of the input in all layers up to the bottleneck:

$$s^i(\mathbf{L}^i) \in \mathbb{R}^{h^i/2 \times w^i/2 \times 4n_c^i} \text{, for } \mathbf{L}^i \in \mathbb{R}^{h^i \times w^i \times n_c^i} \tag{3}$$

$$u^i(\mathbf{L}^i) \in \mathbb{R}^{2h^i \times 2w^i \times n_c^i/4} \text{, for } \mathbf{L}^i \in \mathbb{R}^{h^i \times w^i \times n_c^i} \tag{4}$$

In this sense, the bottleneck is the only moving part in our experiments. The resulting models have a number of parameters ranging from $\sim$ 50M to 90M, depending on the bottleneck shape. The variants with smaller bottleneck area have more parameters due to the large number of channels needed to achieve the same relative volume to the input as variants with a bigger area.

## 2.3 DATASETS

To increase the robustness of our study, we conducted experiments on three different datasets:

**Chess** The first dataset is a collection of synthetic images showing chess positions [1]. The images are generated by randomly sampling the board and piece icon style and the number of pieces present on the board. Each image comes with a string of characters describing the position in FEN notation. We convert this description into binary labels that indicate each possible piece's presence, excluding kings, as they are always on the board. In total, the dataset consists of 100,000 $400{\times}400$ pixel images. To keep the training time within acceptable bounds, we resized all images to be $96{\times}96$ pixels.

**CelebA** Our second dataset is the CelebA faces dataset Liu et al. (2015). This dataset is a collection of 202,600 images showing celebrity faces, each with a 40-dimensional attribute vector (attributes such as smiling/not smiling, male/female). For our purposes, we resized the images to be $96{\times}96$ pixels. The original size was $178{\times}218$ pixels.

**STL-10** For our last dataset, we picked STL-10 Coates et al. (2011). This dataset consists of $96{\times}96$ pixel natural images and is divided into three splits: 5,000 training images (10 classes), 8,000 test images (10 classes), 100,000 unlabeled images. The unlabeled images also include objects that are not covered by the ten classes in the training and test splits.

# 3 EXPERIMENTS

## 3.1 AUTOENCODER TRAINING

The first experiment we conducted, and which forms the basis for all subsequent experiments, consists of training of autoencoders with varying bottleneck shapes and observing the dynamics of their training and test losses. This experiment probes the relative importance of area versus depth in the bottleneck. Additionally, it provides insight into how and when our models overfit. We also

---

[1]https://www.kaggle.com/koryakinp/chess-positions

tested the hypothesis that autoencoders learn to "copy" the input if there is no bottleneck (i.e., the volume of activations relative to input is always 1). To be able to observe overfitting behavior, we used only the first 10,000 images for training and the last 2,000 images for testing in each dataset. For STL-10, we used images from the unlabeled split.

Our CAE model, introduced in Section 2.2, yields bottleneck area $A_0$. We obtain two additional bottleneck areas $A_i$, $i \in \{1, 2\}$ by changing the stride in the last $i$ strided convolution layers in the encoder from 2 to 1.

$$A_i = \frac{A_{in}}{4^{n_l - i}} \qquad i \in \{0, 1, 2\}, \ n_l = 5 \tag{5}$$

In this equation, $n_l = 5$ is the number of strided convolution layers in the vanilla network, and $A_{in}$ is the area (height $\times$ width) of the images in the dataset. The figures in Section 4, refer to $A_0$, $A_1$, $A_2$ as S, M, L (small, medium, large), which we believe to be more intuitive.

For each area setting we then fix four levels of compression $c_j \in \{^1/_{64}, ^1/_{16}, ^1/_4, 1\}$ and calculate the necessary number of channels $n_{c_j}$ according to

$$n_{c_j} = \frac{c_j A_{in} n_{c_{in}}}{A_i} \qquad i \in \{0, 1, 2\}, \ j \in \{1, 2, 3, 4\} \tag{6}$$

Here, $n_{c_{in}}$ is the number of channels in the input image. When presenting our results, we use the levels of compression, rather than the number of channels, as the latter changes based on the chosen area for the same level of compression.

All resulting autoencoders have the same number of parameters in all layers except the convolutional layer and residual block directly preceding and following the bottleneck. We use mean squared error (MSE) between reconstruction and input as our loss function. After initializing all models with the same seed, we train each for 1,000 epochs, computing the test error after every epoch. We repeat this process for three different seeds on each dataset. Training a single model for 1000 epochs took approximately two days on a Nvidia RTX 2080Ti using half-precision.

## 3.2 Scaling with Dataset Size

As described above, we limited the training data to 10,000 samples for the training of our CAEs. To estimate the effect of the training data amount has on CAE training, we train CAE models with six different amounts of training data. For this experiment, we use the CelebA dataset and four of the twelve models from the first experiment (S 1/64, S 1/64, M 1/16, M 1/16). We train the models on the first 1% (2k samples), 5% (10k samples), 10% (20, samples), 25% (50k samples), 50% (100k samples), and 95% (190k samples) of the samples in the dataset while reserving the last 5% for testing. Each model trains for 35,000 iterations, which translates to roughly half of the compute each model received in the first experiment. For each split and model, we use three random seeds, which are different from the ones used in Section 3.1.

## 3.3 Knowledge Transfer

Another goal of our investigation is to estimate the effect of the latent code shape on transferability. Here, we train a logistic regression model on latent codes to predict the corresponding labels for each dataset. Since logistic regression can only learn linear decision boundaries, this approach allows us to catch a glimpse of the sort of knowledge present in the latent code and its linear separability. Furthermore, this serves as another test for the "copying" hypothesis. If the encoder has indeed learned to copy the input, the results of the logistic regression will be the same for the latent codes and the input images. We perform this experiment separately for data that the CAEs saw during training and unseen data. In the first step, we export latent codes for all 108 trained models. From the Chess and CelebA datasets, we use the 10,000 samples in the training data and another 10,000 unseen images from the end of the dataset. Since we trained CAEs on samples from the unlabeled split of STL-10, there are no labels available for the training data. Consequently, we only use the 8,000 images in the STL-10 test split for classification.

We train the classifiers with a one-cycle policy Smith (2018) and early stopping for a maximum of 100 epochs. For each classifier, we use 60% of data for training, 20% as a validation set for early stopping, and 20% for final testing. Additionally, we train each classifier three times on different

seeds and different samples in the splits. Besides, we also train models directly on the image data for every dataset to serve as a baseline for comparison (also with three seeds).

## 3.4 OUTLIER DETECTION

Since CAEs are widely used in reconstruction-based outlier detection, we investigate the impact of the bottleneck shape in this setting as well. To this end, we used the models trained on the CelebA (all configurations and all three seeds) data calculated the reconstruction loss of 2000 samples from the CelebA and STL-10 datasets each. We then clip the losses to 1 and calculate the area under the ROC curve as an indicator for the effectiveness of the AE as an outlier detector.

## 3.5 AUTOENCODER
INTERPOLATION AND ITERATION

To get a more holistic (but qualitative) impression of the differences between the different bottleneck configurations we employ methods for visualizing different aspects of the latent space learned by the models on the CelebA data. The first such method is latent-space interpolation. Here, we pick pairs of test samples from the CelebA dataset and encode them. Once we have the latent codes, we perform liner interpolation between them, reconstructing the intermediates along the way. Visually

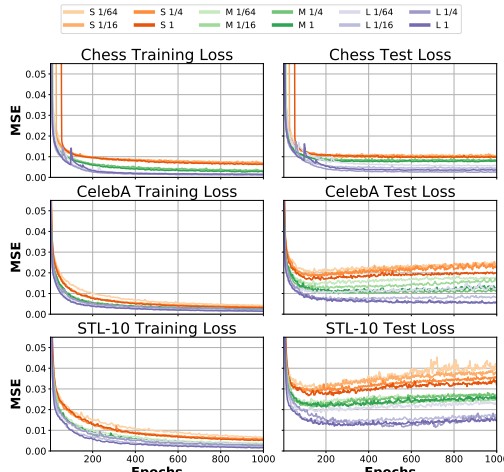

Figure 1: Loss plots for the three datasets. Every bottleneck configuration is shown as a distinct line. Each line is the average over three runs. Configurations that have a common bottleneck area share the same color. Color intensity represents the bottleneck depth (darker = more channels)

inspecting the resulting intermediate reconstructions is a good method for discovering qualitative differences that hard to capture using metrics Oring et al. (2020).

The other method we use is iteration of the CAE. Starting from a test sample we repeatedly feed the reconstructed image back into the autoencoder. Bengio et al. (2013b) showed that this can be thought of as walking along the latent space manifold learned by the model. In our experiment we iterate over each model 100 times. Compared to the interpolation, this method samples different portions of the latent space, as the process is not directed.

# 4 RESULTS AND DISCUSSION

## 4.1 AUTOENCODER TRAINING

**Generalization Improves with Bottleneck Area** We observe that bottleneck shape critically affects generalization (Figure 1). While bottleneck depth has almost no effect on generalization, increasing the bottleneck area vastly improves generalization and enables the models to be trained longer before overfitting occurs. To measure the strength of this effect, we looked at the minimal test error achieved by each model, and the epoch at which this minimum occurred (Figure 2). Models with a larger bottleneck area have a much lower test error than ones with a small area. Additionally, their test error curves reach their minima at later epochs, meaning that such models start overfitting later in the training cycle. The differences in the minimum test error were statistically significant for the different bottleneck areas with p-values below 0.001, according to a Wilcoxon rank-sum test (Table 1 in the Appendix). This finding is surprising, given the hypothesis that only the total amount of neurons matters. The better generalization of models with a larger bottleneck area is also clearly visible when inspecting reconstructions of test samples. We have added figures showing the reconstructions of random samples in the Appendix.

This effect is relevant for many applications of CAEs. For instance, if compression or restoration is the goal, a larger bottleneck area is helpful as we demonstrate with Figures B.1, B.2 and B.3 in the supplementary material. On the other hand, if the CAE is intended for outlier detection, a smaller

bottleneck area is preferable, as this increases the difference in reconstruction error between valid samples and outliers. To show this, we have plotted the difference in reconstruction error for CAEs trained on CelebA data when reconstructing unseen samples from CelebA and STL-10 in Figure C.1

**Training Speed Increases with Bottleneck Area** Although all bottleneck configurations result in similar final training errors, a bigger bottleneck area accelerates the model training. This effect is apparent in the loss curves in Figure 1. Loss curves from models with a bigger bottleneck area reach lower training and test error sooner than their counterparts. We have quantified the effect by looking at the first epoch to reach a training error below 0.01 (Figure 2). We find that only the bottleneck area is strongly correlated with convergence speed, while bottleneck depth does not seem to affect it. A Wilcoxon rank-sum test shows that the epoch at which the training loss falls beneath 0.01 is significantly different for the bottleneck areas, with all p-values being below 0.001, except in two cases, where they were 0.02 and 0.08. (Table 1 in the Appendix). The transition from a small to a medium area yielded the largest speed up in our experiments.

**CAEs without Bottleneck Do Not Learn an Identity Mapping** CAEs, where the total number of neurons in the bottleneck is the same as the number of pixels in the input, do not show signs of simply copying images. If the CAEs would indeed copy images, the test error would go to zero, yet we do not observe this case in any of the datasets.

What is more, these complete CAEs follow the same pattern as the under-complete ones and often converge to similar values. In essence, it suggests that even complete CAEs learn abstractions from data, and raises the question: What prevents the CAE from simply copying its input? We believe that the answer to this question could potentially lead to new autoencoder designs that exploit this limitation to learn better representations. Hence, we argue that it is an exciting direction for future research. Additionally, the trends we derive from our results suggest that this finding likely extends to over-complete CAEs as well. However, experiments with over-complete CAEs are required to test this intuition.

### 4.2 SCALING WITH DATASET SIZE

We present the results of the dataset scaling experiment from Section 3.2 in Fig. 3. Training with more data unsurprisingly results in better generalization but with diminishing returns. When using more than 25% of the full datasets ($\approx$ 50,000 samples), the test error does not improve significantly, as it converges to the training error.

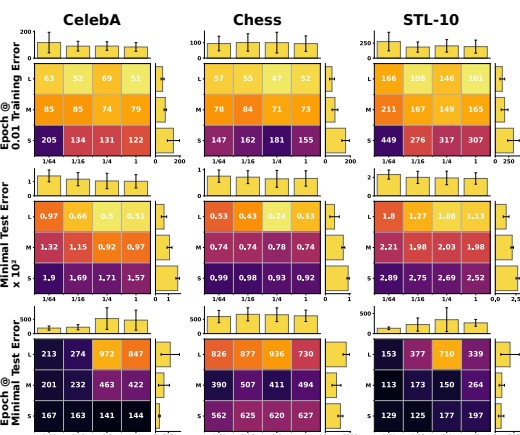

Figure 2: Plots showing the first epoch to fall beaneath 0.01 training error, the minimal test error and the epoch at which the model achieved it (see Section 4.1). Columns represent the results for each dataset. Each number is the average over three runs. Marginals show the mean and standard deviation of the row / column. More desirable values have brighter colors.

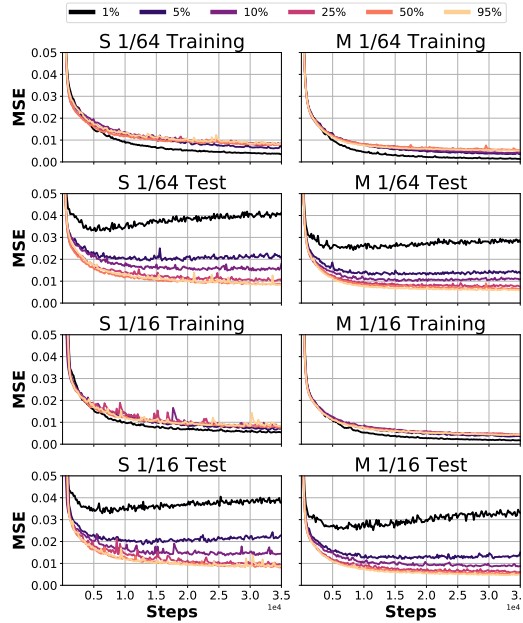

Figure 3: Loss plots for the scaling experiment (see Section 3.2) averaged over three runs. Each quadrant shows training and test losses from one model. Colors correspond to the fraction of data used for training, as shown in the legend at the top. Brighter means more training data.

These models did not plateau after 35,000 training iterations, meaning that differences could arise if we trained the models longer.

We can confirm our observation from Section 4.1 that a bigger bottleneck area results in faster training. As can be seen from the scaling experiment, this effect is independent of the training set size.

The effect of the bottleneck area on generalization, which we discovered in the first experiment, is present for all fractions of the training data. Additionally, we find the decrease in test error to be more pronounced if we train the model on fewer samples.

### 4.3 KNOWLEDGE TRANSFER

If we look at the results of our knowledge transfer experiments (Fig. 4), we find further evidence that contradicts the identity mapping hypothesis. Although the loss curves and reconstructions already indicate that the CAE does not copy its input, the possibility remains that the encoder distributes the input pixels along the channels, but the decoder is unable to reassemble the image. Here, we see that the results from the linear model trained on latent codes of complete CAEs perform better than those trained directly on the inputs (marked "baseline" in the figure). As such, it is implausible to assume that the encoder copied the input to the bottleneck.

Overall, we find that bottleneck shape only slightly affects knowledge transfer with linear classifiers. Interestingly, the number of channels in the code seems to improve the performance of the classification slightly. It is not clear, however, whether this effect is due to the structure of the codes or their high dimensionality. Perhaps projecting the representations to have the same dimensionality (using PCA or UMAP) before classification could answer this question. Additionally, we find that the classifiers' performance is almost the same on seen and unseen data. Given the differences in overfitting between the different models, we would expect otherwise. This may suggest that encoder and decoder exhibit different degrees of overfitting. One possible explanation would be that overfitting happens mostly in the decoder, while the encoder retains most of its generality. We believe that this question warrants further investigation, especially in light of the widespread use of transfer learning methods.

### 4.4 OUTLIER DETECTION

Calculating the difference between the average MSE in both the CelebA and STL-10 samples shows that models with a larger bottleneck area

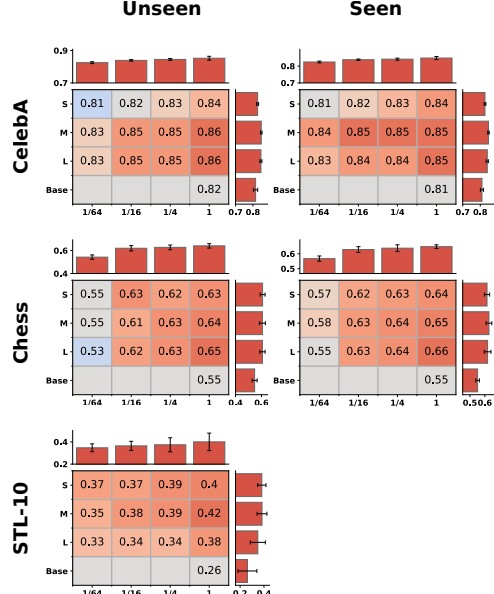

Figure 4: Results linear probes. AUROC for Chess and CelebA, macro F1 for STL-10. Marginals show the mean and standard deviation. Red signifies improvement over baseline.

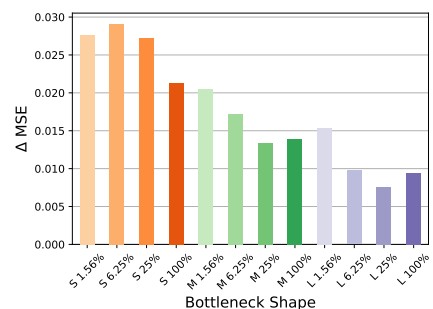

Figure 5: Difference in the average MSE on both datasets in the outlier detection experiment.

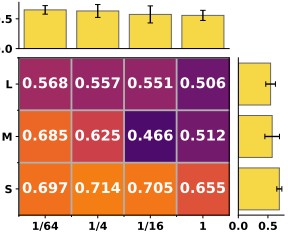

Figure 6: AUROC organized into a grid of area x number of channels

perform significantly better due to their improved generalization capabilities. This poses an issue for reconstruction-based outlier detection however, as can be seen in the results for the calculated

AUROC values (see Fig. 6). In conclusion one can observe, that for outlier detection tasks a smaller bottleneck area is preferable. It appears that the outlier detection performance also decreases with increasing number of channels, although this effect is much weaker and not as clear.

### 4.5 AUTOENCODER INTERPOLATION AND ITERATION

Looking at the results of the interpolation and iteration experiments (see Figures 8 and 7) we find that smaller the bottleneck areas encourage the learning of "concepts". We can see this e.g. in the iteration results, where the models with the smallest bottleneck area still show faces after 100 iterations, while the images of the larger ones dissolve into indistinguishable patterns. Similarly, the iteration from the larger-area models seem to more closely resemble aplha-blending, whereas the models with smaller area show a gradual morphological change of their contents.

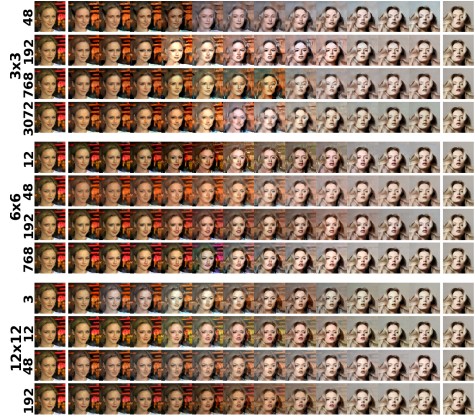

Figure 7: Results of the autoencoder interpolation. Images are grouped by area and increasing with number of channels.

## 5 CONCLUSION

In this paper, we presented the findings of our in-depth investigation of the CAE bottleneck. We could not confirm the intuitive assumption that the total number of neurons sufficiently characterizes the CAE bottleneck. We demonstrate that the height and width of the feature maps in the bottleneck define its tightness, while the number of channels plays a secondary role. Larger bottleneck area (i.e., height × width) is also critical in achieving lower test errors, while simultaneously speeding up training. These insights are directly transferable to the two main areas of application for CAEs, outlier detection and compression/denoising: In the case of outlier detection, the model should yield a high reconstruction error on out-of-distribution samples. Using smaller bottleneck sizes to limit generalization proves useful in this scenario. On the other hand, compression and denoising tasks seek to preserve image details while reducing file size and discarding unnecessary information, respectively. In this case, a bigger bottleneck size is preferable, as it increases reconstruction quality at the same level of compression. Furthermore, we could not confirm the hypothesis that complete CAE (i. e., CAEs with the same number of neurons in the bottleneck as pixels in the input) will learn an identity mapping. On the contrary, even complete CAEs appear to follow the same bottleneck size dynamics, as stated above.

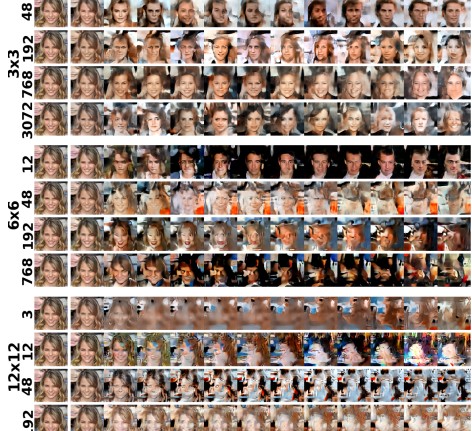

Figure 8: Results of the iteration experiment. Each column shows the reconstruction after 8 iterations.

In knowledge transfer experiments, we have also shown that CAEs that overfit retain good predictive power in the latent codes, even on unseen samples. An interesting question that follows from these experiments is how overfitting manifests itself in CAEs. Does it occur mainly in the encoder or decoder or equally in both?

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
