# OpenReview forum: "Walking the Tightrope: An Investigation of the Convolutional Autoencoder Bottleneck"
_ICLR.cc/2023/Conference — Submitted to ICLR 2023_

### Official Review · Reviewer_fPDr · 2022-10-24

**Confidence:** 4
**Correctness:** 3
**Technical Novelty And Significance:** 2
**Empirical Novelty And Significance:** 2
**Recommendation:** 5

**Clarity, Quality, Novelty And Reproducibility:**

Clarity: the paper is clearly written.
Quality: I have some concerns with respect to the methodology and scope of the experiments (see Weaknesses above).
Novelty: The specific direction of investigating the bottleneck shape of CAEs in-depth is novel to the best of my knowledge.
Reproducibility: code to reproduce the experimental results has been provided.

**Strength And Weaknesses:**

Strenghts:

- The paper is fairly well-written and easy to follow. It is organized in a logical manner.

- The paper's motivation is clear, it is important to gain better understanding of the trade-offs and design considerations of common deep learning models.

- The paper identifies an interesting question with respect to AE overfitting: is overfitting mostly happening in the decoder only?

Weaknesses:

- I have some concerns with respect to the experiments. The datasets used in the core experiments are small (10k images) with small resolution images (96x96). The paper's findings may not necessarily hold in higher dimensions, however in most applications where the findings may be relevant (medical image denoising, outlier detection) the images are typically of much higher resolution.

- The methodology of the experiments can also be somewhat questionable. Changing the bottleneck layers results in models of different size (50M-90M according to the authors) and potentially very different training dynamics depending on bottleneck size. Therefore, using the same training hyperparameters for all models may not guarantee a fair comparison: larger models may need more time to converge properly, the optimal learning rate for different models may vary. It would be important to see how these parameters were set and whether the comparison is really fair.

- I am not convinced that bottleneck area is necessarily the cause. For instance, the observed phenomenon might be explained as follows: all models are heavily overparameterized due to the small dataset/low-res images, the smallest model (in terms of # of parameters) has the strongest implicit regularization, therefore will provide the best generalization. In case of this paper, the smallest model is the one with the largest bottleneck area.

- I don't see how "bottleneck depth has almost no effect on generalization".  According to Figure 1,  there is a very significant spread between different compression ratios, however typically the higher ratio the better.

- I am unsure what was the purpose of the interpolation experiments, they are not directly referenced and no conclusion has been drawn.

- If the authors' conclusion is correct and in fact larger bottleneck area is better, why did we stop at the largest resolution investigated in the paper? What happens if we do not use any spatial downsampling?

- The paper has only 8 content pages out of 9. Many interesting questions raised by the authors could fill the extra page with additional experimentation.

**Summary Of The Paper:**

This paper empirically investigates the impact of the bottleneck shape of convolutional autoencoders (CAE) on generalization, training speed,  outlier detection and transferability of features. Based on numerical experiments on a CAE, authors conclude that larger spatial resolution in the bottleneck corresponds to better generalization.  Furthermore, they claim that convergence speed increases with larger bottleneck area. They also find that CAEs do not learn to copy the input to the output, even when the bottleneck volume matches the input image volume. Based on their findings, authors recommend large bottleneck area for compression/denoising and small bottleneck area for outlier detection.

**Summary Of The Review:**

Overall, in its current form I would lean towards borderline rejecting this paper.  I have concerns with respect to the experimental methodology and the significance of the findings as it is unclear whether the phenomena hold in higher dimensions and on models deployed in practice.  Furthermore, the provided experimental details are insufficient to determine whether the different models were fairly compared. The paper is definitely investigating interesting questions however, and the direction on better understanding the nature of overfitting of AEs would be an interesting addition to the paper.

---

### Official Review · Reviewer_VXUw · 2022-10-25

**Confidence:** 3
**Correctness:** 4
**Technical Novelty And Significance:** 3
**Empirical Novelty And Significance:** 3
**Recommendation:** 5

**Clarity, Quality, Novelty And Reproducibility:**

The paper was easy to understand and clear in their goals and setup. I found the results to be interesting but I am not an expert on autoencoders and am not familiar with other literature in this space so it is difficult for me to assess the novelty of the work.

**Strength And Weaknesses:**

The paper is clear and focused and I found it to be easy to follow. The study is practically useful and the results are interesting, although I am not an expert on autoencoders and do not know of related work to this study.

My main concern with the paper is the utility of the authors conclusions given the single model configuration that was studied. It is not clear to me that these observations will generalize to a different convolutional autoencoders architectures.

I also found the training time of 1000 epochs to be surprisingly long - roughly an order of magnitude more than is typical for vision datasets like ImageNet, which is much larger than the datasets studied here.

**Summary Of The Paper:**

The authors conduct a study of convolutional auto-encoders to assess the importance of different dimensions of the autoencoder bottleneck on model quality for a range of tasks. They additionally assess whether or not these models learn identity functions when the number of pixels in the input matches the number of features in the bottleneck. Based on their analysis, the authors make practical recommendations for the designing autoencoder architectures.

**Summary Of The Review:**

Overall, the paper is well written and the authors' results are potentially practically useful. My primary concern is the single architecture considered and whether the authors' results will generalize beyond this setup.

---

### Official Review · Reviewer_7HZy · 2022-10-26

**Confidence:** 3
**Correctness:** 3
**Technical Novelty And Significance:** 3
**Empirical Novelty And Significance:** 2
**Recommendation:** 6

**Clarity, Quality, Novelty And Reproducibility:**

This paper is well-motivated. The writing is clear and easy to follow. The analysis seems novel and insightful.

**Strength And Weaknesses:**

Strength:
- The analysis is comprehensive and insightful. The conclusions seem rational.
- Several realistic applications are considered, i.e., image compression, outlier detection, and classification.

Weaknesses:
- I'm not fully convinced by the claim that "a smaller bottleneck area is preferable for outlier detection". From Figure 1 one can observe that a smaller bottleneck area leads to overfitting. The test reconstruction loss of both normal samples and outliers will be large. Why is this beneficial for outlier detection?
- Are there some comparisons against the recently proposed methods of image compression? The improvements on top of state-of-the-art baselines may be strong evidence to demonstrate the practical importance of the conclusions of this paper.

**Summary Of The Paper:**

This paper conducts an analysis of the bottleneck size/channels of the convolutional autoencoder (CAE). The authors find that an increased bottleneck area (i.e., height × width) improves generalization in terms of reconstruction error while also speeding up training. In contrast, decreasing the feature map size will make out-of-distribution samples yield a higher reconstruction error, which may be beneficial for outlier detection.

**Summary Of The Review:**

Personally, I think this is a good paper. Although there are some notable weaknesses, I vote for borderline acceptance.

---

### Official Review · Reviewer_AyQZ · 2022-10-31

**Confidence:** 3
**Correctness:** 2
**Technical Novelty And Significance:** 2
**Empirical Novelty And Significance:** 2
**Recommendation:** 3

**Clarity, Quality, Novelty And Reproducibility:**

The paper is easy to reproduce and included code, results, models and dataset in a anonymous github.

**Strength And Weaknesses:**

The paper findings are highly experimental, which is not a defect in itself, but creates an expectation for robust experimental settings that reflect the hypotheses investigated. In this sense, the major weakness of the paper is the lack of extensive experiments to support the recommendations and conclusions made and/or the lack or real case study. In the same direction, the paper mentions that the insights are 'directly transferable' to outlier detection and compression/denoising, but no evidence of such is given.

All references pointed are dated a few years ago with a single reference from 2020. Again here, using older references is not a defect in itself, but should not be the case in the lines where authors indicate 'currently' state of the art/on going research directions (Intro- first paragraph). This indicates that the paper contextualization is outdated as it contrasts with the velocity of new findings in the main application areas tackled in the paper (compression and outlier detection).


**Summary Of The Paper:**

The paper investigates the convolutional autoencoder (CAE) bottleneck, more specifically on the impact of the bottleneck shape. Empirically, under the experimental settings chosen, it observes that increased bottleneck spatial resolution (i.e., height × width) improves generalization in terms of reconstruction error while speeding up training, while the chosen number of channels in the bottleneck is of secondary importance.
The ablated model consists of a convolutional auto encoder with five conv. layers in the encoder and another 5 in the decoder, such that given an input with dimension 96x96, produces a bottleneck with spatial resolution of 3x3, and kept the 'volume' (height × width × channels) constant across experiments.
The models with different spatial/depth composition is trained on 3 datasets: Chess (chess positions), CelebA (40-attribute vestor) and STL-10 (10 classes + unlabeled), resized to 96x96 pixels.
The results presented are comparison of training and test loss curves for the 3 datasets, and bottlenecks scaled by 1, 1/4 and 1/64.
It conjectures that the observations can be used to recommendations on outlier detection and compression/denoising, suggesting to increase the feature map spatial resolution for image compression but reduce it for reconstruction based outlier detection.


**Summary Of The Review:**

The paper is clear, but does not present a robust experimental setting to sustain the conclusions/recommendations made. In that sense, it is hard to evaluate its impact and validity outside the experiments performed into real  use-cases of CAE.

---

### Decision · Program_Chairs · 2023-01-20

**Decision:**

Reject

**Justification For Why Not Higher Score:**

All but one reviewer are against acceptance on the grounds that they doubt the paper's conclusions would generalize beyond the experimental setting considered. The authors have not responded to the reviews.

**Justification For Why Not Lower Score:**

N/A

**Metareview: Summary, Strengths And Weaknesses:**

The submission looks into the impact of the bottleneck structure in convolutional autoencoders on generalization. Based on their empirical observations, the authors recommend to increase the bottleneck feature map's spatial resolution for image compression and to decrease it for outlier detection.

Reviewers noted the submission's writing clarity but expressed doubts that the conclusions drawn in the paper would generalize beyond the experimental setting considered. The authors have not responded to the reviews.

I recommend rejection.

**Summary Of Ac-Reviewer Meeting:**

N/A